# Functional Thyroid Organoids—Powerful Stem Cell-Derived Models in Basic and Translational Research

**DOI:** 10.3390/biom15050747

**Published:** 2025-05-21

**Authors:** Meghna Parakkal Shankar, Alessandra Boggian, Daniela Aparicio-Quiñonez, Sami Djerbib, Eduardo Rios-Morris, Sabine Costagliola, Mírian Romitti

**Affiliations:** Institute of Jacques-Dumont Interdisciplinary Research in Molecular Human Biology (Jacques-Dumont IRIBHM), Université Libre de Bruxelles, 1070 Brussels, Belgium; meghna.parakkal.shankar@ulb.be (M.P.S.); alessandra.boggian@ulb.be (A.B.); daniela.aparicio.quinonez@ulb.be (D.A.-Q.); sami.djerbib@ulb.be (S.D.); eduardo.andres.rios.morris@ulb.be (E.R.-M.); sabine.costagliola@ulb.be (S.C.)

**Keywords:** thyroid development, thyroid organoids, stem cells, thyroid diseases, disease modeling, congenital hypothyroidism, endocrine disruptors, thyroid transplantation, regenerative therapy

## Abstract

Thyroid organoids, three-dimensional in vitro models derived from stem cells, have emerged as a powerful tool for studying thyroid development, function, and disease mechanisms. These organoids recapitulate the key aspects of the thyroid gland, including the follicular structure, hormone production, and response to stimuli such as to the thyroid-stimulating hormone (TSH). Recent advances in thyroid organoid technology have established the basis for the modeling of development and thyroid diseases, including congenital hypothyroidism (CH), autoimmune conditions like Graves’ disease and Hashimoto’s thyroiditis, and other thyroid-related disorders. By utilizing pluripotent stem cells (PSCs) and adult tissue, researchers have generated organoid models suitable for dissecting the mechanisms associated with thyroid development while mimicking the genetic, functional, and inflammatory characteristics of thyroid diseases. Additionally, thyroid organoids offer the potential for personalized medicine by providing a platform to test therapies in a more clinically relevant context. This review highlights the recent progress in thyroid organoid generation, discusses their applications in dissecting the thyroid development mechanisms and disease modeling, and explores their potential for advancing our understanding of the thyroid physiology and pathology. Furthermore, we address the challenges and future directions in the optimization and use of thyroid organoids in translational research.

## 1. Introduction

The thyroid gland is the largest endocrine gland in the human body and the first endocrine organ to develop during human embryogenesis. Its development initiates in the third gestational week (GW) and becomes functional by the 11th GW [1]. The thyroid gland arises from the anterior-most region of the endoderm-derived gut tube. It is first characterized by the co-expression of *NKX2-1*, *PAX8*, *HHEX*, and *FOXE1* transcription factors essential for deriving thyroid-specific progenitors [2,3,4]. These originate from a midline structure in the pharyngeal floor known as the thyroid diverticulum. This early step in thyroid development, termed specification, is thought to rely on morphogenetic signals from adjacent tissues such as the cardiac mesoderm, which has been suggested to be the source of the inductive signaling required for thyroid specification [5,6,7]. The thyroid bud forms near the apical pole of the aortic sac and later relocates to its final position at the base of the neck near the trachea. During this process, thyrocytes proliferate and mature, and the gland assumes its species-specific shape and organization [3].

Structurally, the thyroid gland is composed of two primary endoderm-derived cell types: (1) epithelial follicular cells, also known as thyrocytes, which synthesize triiodothyronine (T3) and thyroxine (T4), and (2) parafollicular cells (C cells), which secrete calcitonin, a hormone involved in calcium metabolism. The functional unit of the thyroid is the thyroid follicle, a spherical structure composed of a monolayer of thyrocytes encircling a central lumen filled with colloids. This protein-rich substance, the colloid, is a storage reservoir for thyroid hormones (THs). The follicle and the adjacent microvasculature form the basic morphological and functional thyroid unit, known as the angio-follicular unit (AFU), with a crucial role in thyroid homeostasis, growth, development, and function [8]. The thyroid AFUs also ensure the efficient delivery of hormones to distant target organs. The thyroid plays a critical role in regulating growth, development, neural differentiation, and metabolism, through the production of T3 and T4. These hormones are essential for the embryonic and post-embryonic development across vertebrates [1,9].

The molecular mechanisms underlying the distinct stages of thyroid development have been partially elucidated through in vivo studies, thereby providing the foundation for the generation of thyroid organoid models (previously reviewed) [1,3,5,9,10]. The stages of thyroid gland formation are largely conserved among vertebrates. However, the interspecies differences and limitations of mammalian models challenge the studies on the early developmental mechanisms and modeling of human diseases [11].

In this review, we explore the recent advances in the generation of thyroid organoid models and examine their impact on understanding thyroid developmental biology and disease. Additionally, we discuss the potential applications of these systems in translational medicine and the challenges that need to be addressed for their broader implementation.

## 2. Organoids: An Advanced Tool to Dissect Thyroid Development and Model Diseases

Organoids are three-dimensional (3D) culture systems derived from primary tissues containing a stem cell or a progenitor compartment, embryonic stem cells (ESCs), or induced pluripotent stem cells (iPSCs). They replicate the organ architecture and function, maintaining their ability for self-renewal and self-organization [11,12,13]. Although most organoids lack the full diversity of cell types found in tissues, they allow the study of cell–cell interactions in a controlled microenvironment [14]. By manipulating niche components and signaling pathways, organoids enable the study of cellular or tissue interactions in 3D co-culture, offering a more physiologically relevant platform than traditional monolayer culture models [15]. Moreover, organoids hold promise for generating isogenic adult tissues for transplantation in regenerative medicine [14,16]. The short-term 2D cultures of thyroid follicular cells (TFCs) have long been used to study the adult thyroid physiology but could not form follicular structures. Suspension cultures allowed 3D follicular organization but did not support long-term growth [15]. Past efforts using primary adult tissues (rat, human, or pig thyroid follicles) cultured in extracellular matrices successfully produced thyroglobulin (TG) and thyroid maturation factors, but these structures did not qualify as organoids [17,18].

The groundbreaking work of Hans Clevers’ team in 2009 established the foundation for stem cell-derived organoids pioneering their use as models for studying the development of the different organs. By isolating Lgr5+ intestinal stem cells capable of differentiating and organizing into intestinal epithelium, Clevers’ group generated the first intestinal organoids [19]. On the other hand, the approaches to generate thyroid organoids were based on the specification of PSCs into thyroid progenitors and further differentiation. This is notably due to two fundamental properties of the thyroid gland: low cell turnover and the lack of resident stem cells [20,21,22,23]. However, organoids have been derived from fetal and adult thyroid tissues, suggesting the presence of a progenitor compartment [19,21,23]. In the next section, we will explore the distinct organoid methods available and how they replicate the distinct stages of thyroid development and adult conditions in vitro.

For this review, we performed a systematic literature search using PubMed and Scopus. The search strategy included terms such as “thyroid organoids”, “thyroid in vitro models”, “endoderm-derived organoids”, “thyroid transplantation”, “thyroid graft”, “thyroid hormone synthesis in vitro”, “thyroid models”, and “endocrine organoids”. After an initial screening based on abstract content, we included all peer-reviewed studies published to date describing the generation and/or application of pluripotent stem cell (PSC)- or tissue-derived thyroid organoids to study thyroid development, hormone synthesis, and diseases (except thyroid cancer). Only English-language articles providing the relevant experimental data were selected.

### 2.1. Existing Models of Thyroid Organoids

Thyroid organoids have been derived from PSCs or tissue-specific progenitor cells. The latter are generated by the dissociation of a thyroid tissue sample (from biopsies, surgical specimens, or fetal material) and cultured in 3D [24]. Organoids generated from primary tissue often represent a better alternative to modeling the thyroid function in vitro, as primary thyrocytes are fully differentiated and express the necessary suite of genes for hormone synthesis [25]. Many studies have suggested the presence of stem cell-like populations in the thyroid of mice [20] and humans [26], due to the ability of adult tissue to form functional organoids. For example, Lan et al. (2007) have reported the identification of a population of ABCG2+ and OCT4+ cells responsive to thyroid-stimulating hormone (TSH), fueling the search for thyroid stem cells [26]. However, there is still no definitive evidence for the existence of these cells [20,21]. Even elaborate techniques like single-cell RNA sequencing (scRNA-seq), which focus on the human thyroid gland, have not been able to identify or isolate such a cell population [21]. Although thyroid organoids derived from primary tissue have been used to model physiology and functionality, they are not well suited for studying early thyroid development; also, healthy primary thyroid tissue is difficult to obtain and hence is a limited resource. Primary tissue is also affected by the genetic variability from individual to individual, which would impact and bias in vitro findings. On the other hand, PSC-derived organoids, produced from an unlimited source, recapitulate the developmental steps, and offer the potential for transplantation and studying the molecular basis of thyroid congenital diseases. They can be generated via transcription factor overexpression (forward programming) or stepwise differentiation protocols (directed differentiation) using growth factors or inhibitors to try to reconstitute, in vitro, the developmental cues that govern embryonic development (Table 1). It is important to note that in some cases, PSC-derived organoids can contain cell lineages from the three germ layers, owing to the steps that mimic gastrulation. The presence of other cell types could influence the proper maturation of the organoids as a result of the essential cellular interactions involved in early development and maturation [27,28]. In the following section, we will dissect and discuss in detail each step for the generation of PSC-derived thyroid organoids.

### 2.2. From Endoderm to Thyroid Specification

The PSCs need specific developmental cues to exit pluripotency, acquire distinct cell fates, and form the blueprint of the three germ layers [38]. Keller et al. (1995) suggested that these developmental processes could be replicated in vitro using specific pathway activators or inhibitors, applied in defined combinations, sequences, dosages, and timings, thus forming the basis of directed differentiation protocols [39].

A key starting point for PSC-derived organoid formation in many protocols, whether directed or forward programmed, is the generation of self-assembling structures analogous to post-implantation embryonic tissues, known as embryoid bodies (EBs). Since the thyroid gland originates from the endoderm, most differentiation protocols for thyroid organoids begin by generating definitive endoderm (DE), the innermost germ layer of the gastrulating embryo [40]. Gastrulation, in vivo, is regulated spatially and temporally by several signaling pathways such as the wingless-type MMTV integration site family (WNT), Transforming Growth Factor-beta (TGF-beta), Nodal, Bone Morphogenic Protein (BMP), and Fibroblast Growth Factor (FGF). The reciprocal interactions between the germ layers occur through secreted factors. For instance, mesodermal tissue secretes FGF, Retinoic Acid (RA), BMP, and WNT, which pattern the endoderm [41]. Morphogen gradients and thresholds can induce specification [42].

Lin et al. (2003) demonstrated that high Nodal/TGF-beta signaling induces endodermal genes like *Foxa2*, while lower levels produce mesodermal cells expressing markers such as Brachyury. Activin A (AA), also a member of the TGF-beta superfamily, is commonly used to induce DE formation in vitro [43].

Studies using mouse ESCs (mESCs) [43,44,45] provided the proof of concept that thyroid follicular cells could be generated from PSCs in vitro. These studies primarily employed embryoid bodies for mESC differentiation towards thyroid fate, following the withdrawal of leukemia inhibitory factor (LIF), initially required to maintain the undifferentiated status of mESCs [46]. The generated EBs were subsequently treated with Activin A (AA) to induce definitive endoderm (DE) differentiation, characterized by *Sox17* and *Foxa2* expression, and TSH for the thyroid fate [45]. Studies using zebrafish, *Xenopus*, and chicks demonstrated that BMP signaling posteriorizes the endoderm during early development [47,48]. Since thyroid progenitors originate from the most anterior portion of the endoderm layer, also known as Anterior Foregut Endoderm (AFE), studies often use BMP and TGF-beta signaling inhibitors, such as Noggin/dorsomorphin and SB-431542, respectively, to induce AFE [29,30,49,50].

One of the primary challenges in efficiently differentiating PSC-derived endoderm into thyroid follicular cells is the incomplete understanding of the pathways regulating the early thyroid development [29]. Taking advantage of the molecular characterization of thyroid progenitors provided by in vivo studies, in 2012, our team applied transient overexpression (Doxycycline (Dox)-inducible system) of the thyroid transcription factors, *Nkx2-1* and *Pax8*, in mESCs to “force” thyroid specification [31]. In vitro, *Nkx2-1* and *Pax8* co-expression is also shown to sufficiently drive the differentiation toward the thyroid lineage, likely through an autoregulatory mechanism that activates their endogenous expression [1,32]. Briefly, mESC-derived EBs were embedded in Matrigel (3D culture), and *Nkx2-1* and *Pax8* expression was promoted by Dox (3 days). After Dox removal, the cells showed the endogenous activation of the *Nkx2-1*, *Pax8*, *Foxe1*, and *Hhex* genes, validating the generation of thyroid progenitors. Using a similar strategy, we recently demonstrated that EBs derived from human ESCs committed to endoderm fate by adding AA, and upon forward programming relying on *NKX2-1* and *PAX8* overexpression, also generates thyroid progenitors [28].

In contrast, using a directed differentiation approach, Longmire et al. (2012) showed the derivation of thyroid and lung progenitors from mESCs by modulating the signaling pathways using activators and inhibitors [29]. mESC-derived AFE cells treated with BMP4 and FGF2 were shown to acquire a thyroid fate [29,30], achieving 16% of the cells expressing Nkx2-1 by day 14 of the differentiation. The novelty of this study is that they were able to capture and produce the primordial lung and thyroid endodermal progenitors from PSCs without the overexpression of essential TFs. Capturing this progenitor state is crucial, as its transient nature makes it difficult to study in vivo. In another study, the same group identified the conserved roles for BMP and FGF signaling in specifying the thyroid lineage from the AFE using mESCs and hiPSCs [29]. Among the cells expressing Nkx2-1 generated by this protocol, 4.9% of the cells also expressed Pax8. In three consecutive studies, Ma et al. (2013, 2015a, and 2015b) showcase the successful differentiation of mESCs, iPSCs, and hESCs, respectively, into differentiated thyrocytes by forward programming and stimulation with AA and TSH [32,33,34]. Similarly, combining forward and directed differentiation, Dame et al. (2017) used mESCs to show the importance of stage-specific *Nkx2-1* overexpression, which converts AFE into thyroid epithelium [35]. Interestingly, the inductive effect of *Nkx2-1* overexpression depends on the competence of the AFE to convert to thyroid epithelium, influenced by the duration of the anteriorization, level of *Foxa2*, and BMP4-FGF2 signaling. This protocol generates 29.3% Nkx2-1+ cells of which 5% co-express Pax8. Recently, Undeutsch et al. (2024) developed a protocol for the efficient generation of thyroid cells from human iPSCs [36]. By shortening the duration of DE induction followed by AFE promotion, they showed that endodermal cells are more prone to becoming thyroid progenitors. These cells were then exposed to BMP and FGF for a 10-day period resulting in around 35.4% of the total cells co-expressing NKX2-1 and PAX8. Some of the parameters obtained from different publications on thyroid organoid production are summarized in Table 1.

As mentioned above, in vivo studies demonstrated the role of BMP and FGF in thyroid development, with the cardiac mesoderm being the potential signaling source [6,7,51]. Insights from scRNA-seq demonstrated the presence of both cardiac and thyroid lineages within mouse and human thyroid organoids [27,28]. The analysis of ligand–receptor interactions showed that cardiovascular cells are an important source of BMP2, BMP4, and FGF2 ligands, whereas thyroid cells express the cognate receptors [51,52]. However, the role of cardiac mesoderm on thyroid cell specification in vitro is yet to be determined.

### 2.3. Thyroid Progenitor Expansion and Differentiation

The next goal to generate functional thyroid organoids is to increase the progenitor pool and provide maturation cues. Progenitor expansion is associated with an increase in cell proliferation which is triggered by activating the cyclic adenosine monophosphate (cAMP) pathway [28]. Transcriptomic analyses of the hESC-derived thyroid organoids also showed an increased expression of the early thyroid markers (*NKX2-1*, *TG*, and TSH receptor (*TSHR*)) upon cAMP treatment, confirming its role in early thyroid differentiation. Even with the significant increase in *TSHR*, which is an indicator of maturation, functional markers such as the sodium/iodide symporter (*NIS*), thyroid peroxidase (*TPO*), and the dual oxidase (*DUOX*) family were not impacted by cAMP treatment and were not sufficient for full thyroid maturation [28]. However, in mESC-derived thyroid organoids, TSH treatment for three weeks facilitated the proliferation, expansion, and full maturation of organoids [31]. Human ESC-derived thyroid organoids, on the other hand, require cAMP for the expansion phase, while the exposure to TSH, dexamethasone, and a TGF-beta inhibitor was required for the maturation of thyroid cells [28]. Inflammation and TGF-beta activation were reported previously to be detrimental to thyroid differentiation [27,28].

Directed differentiation-derived mouse thyroid progenitors [29,30] were shown to give rise to partially mature thyroid organoids in vitro which after xenotransplantation, acquired full maturation and rescued the T4 physiological levels in hypothyroid mice. Despite the similar partial maturation promoted by BMP4 and FGF2 in human organoids in vitro, transplantation was not sufficient to produce functional thyroid organoids [36]. These findings not only highlight the importance of BMP and FGF signaling in thyroid lineage specification but suggest that in vitro conditions may lack the critical factors that are required for complete functional maturation.

### 2.4. Thyroid Hormone Synthesis

THs are produced by fully mature and functioning follicle structures. To achieve full maturation and obtain a functioning follicular structure, all maturation markers and sufficient levels of the transporters need to be expressed.

Among the models described so far, few have been able to produce organoids capable of TH production in vitro [22,23,28,31,35,37,53]. Considering the maturity levels of the starting material, organoids generated from primary tissue have shown thyroid hormone-producing capacity while few PSC-derived thyroid organoids have reached this step because of a lack of full functional maturation.

While PSC-derived thyroid organoids mainly assessed functionality in vitro by showing immunostaining of the TH precursor, iodinated Tg (Tg-I) [31], or T4 [28,35], Ma et al. (2020) evaluated T4 in the culture supernatant in a human iPSC-derived model [37]. They demonstrated that the T4 levels were increased compared to undifferentiated cell cultures while showing similar levels to the ones produced by immortalized Fisher rat thyroid (FRTL-5) cells.

Adult tissue-derived organoids have been employed in multiple studies to show the ability of cells from disaggregated tissue to form functional follicular structures. Ogundipe et al. (2021) used adult thyroid tissue-derived organoids to highlight the differences in thyroid follicular structure-forming potentials between mouse and human adult thyroid-derived cells cultured as organoids [22]. In this study, thyroid organoids from mice and humans were cultured in an expansion media with growth factor supplements adapted from Kurmann et al. (2015). They observed the ability of both murine and human organoids to form thyroid gland-resembling structures with self-renewal capacity. Follicular structures derived in vitro expressed the key transcription factors, NKX2-1 and PAX8, as well as TG accumulation, while T4 staining was detected in 8% of mouse and 21% human thyroid organoids.

Van Der Vaart et al. (2021) used adult thyroid tissue from mice and humans to generate 3D thyroid follicular cell organoids [20]. These organoids were generated from dissociated adult primary thyroid tissue in expansion media supplemented with specific factors with the addition of TSH for human thyrocytes. Organoids derived from different mice and human donors expressed similar levels of thyroid-maturation markers while Free T3 (FT3) quantification at the conditioned media, luminal, and intracellular compartment showed a variability in the T3 levels among donors. However, despite not being directly evaluated by the authors, no clear correlation is observed among the FT3 and *TG, TPO*, and *TSH* expression levels.

Working with single cells coming from minced mouse adult thyroid tissue embedded in Matrigel and incubated in an organoid culture medium containing growth factors and TSH, Saito et al. (2018) developed a mature and functional mouse thyroid organoid model able to synthesize TG, iodide uptake, and FT3 production in vitro under TSH stimulation and iodine treatment [23]. Similarly, Liang et al. (2022) demonstrated that the long-term culture of human fetal thyroid organoids (hFTOs) in a growth factor-supplemented medium recapitulates the normal follicular cell characteristics with Free T4 (FT4) secretion under forskolin stimulation [54].

To mimic the microphysiological environment of the native thyroid gland in vitro and improve the functionality of thyroid organoids, an alternative strategy, the thyroid organ-on-a-chip system was developed, integrating microfluidics, bioengineering, and 3D thyroid organoid cultures [55]. The study successfully generated an organ-on-a-chip using thyroid organoids derived from mESCs [31]. The organoids retained their follicular architecture, and the dynamic perfusion significantly enhanced the proportion of T4-producing follicular structures from around 1% in static cultures to more than 50% under flow conditions. This improvement was not correlated to an increase in the levels of thyroid maturation markers and was most likely due to the continuous supply of nutrients, particularly iodide, and the efficient removal of waste products, simulating the highly vascularized and perfused environment of in vivo thyroid follicles. These findings provide compelling evidence that dynamic culture systems, such as a thyroid organ-on-a-chip, support the maintenance of functional T4 production in vitro.

In conclusion, several organoid models have shown relevant functionality in vitro illustrated by their ability to uptake iodide and produce THs. This underlines the importance of understanding THs synthesis in vivo to then be able to optimize the culture conditions for the efficient functional maturation of thyroid organoids in vitro.

### 2.5. In Vivo Transplantation and Rescue of Hypothyroidism

Replicating the niche required for thyroid functionality in vitro remains a significant challenge. Grafting in vitro cultured organoids into an in vivo environment offers the potential to enhance tissue maturation, a key step to reach functionality. The transplantation of adult tissue and PSC-derived thyroid organoids has shown a remarkable effect on functionality in most cases (Figure 1).

The mESC-derived organoids generated by our group showed consistent T4 production, release of THs into the blood flow (presence of blood vessels within the grafted tissue), and consequently the ability to correct THs plasma levels in thyroid-ablated hypothyroid mice (intraperitoneal ^131^I injection at 5 weeks of age) [31]. This was the first study showing that T4 production subsequently leads to symptomatic recovery, illustrated by normal T4 levels and physiological body temperature. In a following study, Kurmann et al. (2015) transplanted mature mESC-derived organoids generated via directed differentiation, under the KC of hypothyroid mice following thyroid radioiodine ablation. Starting two weeks after transplantation, an increase in the circulating T3/T4 levels was observed with a steady increase until the T3/T4 return to physiological levels by 8 weeks post-transplantation [30].

In 2022, our group used non-obese diabetic (NOD)-SCID mice with RAI-ablated thyroid glands to show that human ESC-derived thyroid organoids could restore the TSH, TH, and thyroid markers’ expression levels after 5 weeks of xenotransplantation under the KC. This model showed high efficiency and a fast effect probably due to the presence of stromal cells within the grafted cell population, showing a better organization of the derived follicles and the presence of blood vessels (from the host) in proximity [28]. Notably, the study showed a more effective recovery of T3 levels compared to T4, which might be compensated for at a later stage post-transplantation, what is still to be evaluated. However, a negative correlation between TSH and T4 was shown, reflecting physiological norms. Even though the expression of the thyroid functional markers, *TG*, *TPO*, *NIS*, and *TSHR*, was comparable to thyroid tissue, no correlation was calculated among the gene expression levels and the TH production, which could indicate whether the level of maturation of the organoids determines their capacity to produce T3 and T4.

Using adult mouse thyroid tissue-derived organoids, Saito et al. (2018) evaluated the functionality in vivo via the transplantation of these organoids under the KC. The graft resulted in a thyroid-like structure capable of iodide uptake 4 weeks after transplantation. However, TH secretion was not evaluated [23]. Using a similar approach, Ogundipe et al. (2021) showed that the transplantation of murine tissue-derived thyroid resulted in 10% of the thyroid follicular structures that produced T4 within 8 weeks, with the proportion increasing to 33% after 17 weeks [22]. On the other hand, after 26 weeks, the grafted human organoids formed follicles expressing T4 in 24% of the structures, with an increasing proportion of T4-producing structures at 29 weeks (30%). The longest delay compared to the previously published work is probably due to the number of transplanted cells and the absence of exogenous TSH in the culture medium during organoid maturation. By transplanting hFTOs into NOD SCIDGamma (NSG) mice, Liang et al. (2022) demonstrated an increase in the T4 levels in grafted mice compared to non-grafted mice after 7 weeks [54].

Undeutsch et al. (2024) transplanted human iPSC-derived TFCs in *Foxn1*^nu/nu^ hypothyroid mice, four weeks after RIA. The graft showed well-vascularized follicle-like structures positive for NKX2-1 and TG, 12 weeks after transplantation [36]. However, no T4 production nor a decrease in TSH in the recipient mice was observed, which the authors speculate to be due to the low levels of *NIS* expression in the organoids.

Overall, transplantation is needed to restore the functionality of an organ, which relies on the interactions among the different components of the tissue: epithelium, parenchyma, mesenchyme, and vasculature [56]. Despite the clear improvement in functionality promoted by thyroid organoids via in vivo transplantation, the methodological differences across models, and the variable outcome assessments, the key factors essential for an effective transplantation are still to be further elucidated. Among the factors potentially critical, the initial status of sample transplantation, as single cells or enriched follicles, could play a role in TH production and the time needed to reach hormonal recovery. In addition, the maturation status of the organoids transplanted as well as the number of cells/structures to be grafted needs to be known. TH production and TSH levels have also been assessed via distinct methodological approaches complicating the comparability of the models’ outcome. The vascularization of grafts is critical for ensuring the delivery of essential nutrients and oxygen required for the survival, removal of the metabolic waste, and in the case of the thyroid, for the TH release (MTC8) and transport to the target tissues. Subsequently, the presence of T4/T3 among transplanted organoids is not enough to ensure TH compensation in vivo, and the assessment of the blood vessels is mandatory to ensure the full potential of grafting organoids as an alternative to compensate for the hypothyroidism in vivo. These insights will aid in improving the efficiency of grafting, promoting full maturation and TH delivery to the target tissues.

## 3. Modeling Thyroid Health and Diseases Using Organoids

As outlined in the preceding chapters, thyroid organoids do not only recapitulate the physiology of the thyroid gland but also illuminate the novel molecular and cellular mechanisms involved in thyroid development, from cell specification to functional maturation. As a result, thyroid organoids have emerged as a powerful tool for modeling thyroid diseases, both congenital and adult-onset conditions, and the alteration of thyroid function due to exposure to specific toxic components, such as endocrine-disrupting chemicals (EDCs). Thyroid function could also be impacted by autoimmune damage, iodine imbalances, congenital conditions, and surgery or radioactive ablation of the thyroid gland, typically used to treat hyperthyroidism or thyroid cancer. Here, we highlight the primary organoid models employed to study congenital thyroid conditions and discuss future directions to address the unexplored aspects of thyroid pathophysiology. It is worth noting that this review will not cover thyroid cancer organoids, a subject that has been extensively recently reviewed elsewhere [57,58].

### 3.1. Modeling Congenital Hypothyroidism with Organoids

CH represents one of the most frequent congenital diseases (1 in 2500–3000 newborns) and is the most frequent preventable cause of mental retardation [59]. Due to the critical role of THs in early brain development, undetected and untreated CH results in severe motor delay, and mental and growth retardation [59,60]. Newborn screening programs have dramatically improved the outcome of affected patients, but new challenges in diagnosis and treatment of CH need to be addressed.

Congenital thyroid disease can arise from defects in any part of the hypothalamic–pituitary–thyroid axis or hormone transport/action [60,61,62]. The most common cause of CH is still iodine deficiency, with a higher incidence in developing countries [63,64]. The second most common form is primary or thyroidal CH, which is subsequently classified into either thyroid dyshormonogenesis or thyroid dysgenesis (TD). Only 15% of CH cases are caused by dyshormonogenesis, defined as disturbances in the mechanisms required for TH synthesis. In these cases, patients usually show goiter (enlarged glands). On the other hand, the remaining 80–85% of all CH cases are due to TD, encompassing all different forms of defective thyroid development: athyreosis—patients with no detectable thyroid but variable levels of TG, hypoplasia-eutopic thyroid but reduced in size and function, hemiagenesis-development limited to only one thyroid lobe with or without the isthmus, and ectopy-thyroid tissues mislocated and presenting varying levels of functionality [2,9,59,60,61,62].

Multiple genes have been implicated in CH, depending on the developmental stage [65,66,67,68,69,70]. However, many of these were identified by Next Generation Sequencing (NGS), and their causative effect or molecular mechanism remains unclear [62]. Altogether, these genes account for only 15% of cases; therefore, elucidating the role of these associated genes, as well as the still unidentified molecular causes of congenital hypothyroidism with thyroid dysgenesis (CHTD), remains a challenge. TD is known to show a clear female prevalence, epidemiological reports claim varying incidences among ethnic groups, and CH incidence in consanguineous marriages is more common, all suggesting a possible common genetic implication [65,71]. Nevertheless, since CHTD occurs almost exclusively in a sporadic mode, and most monozygotic twin pairs are discordant for CHTD, a non-classical Mendelian mechanism of inheritance can be expected.

Although organoid systems have been developed in other fields to study pathological mutations, thyroid organoids are only starting to be adopted. The first reported functional organoids model derived from mESCs by Antonica et al. (2012) has been employed to assess the impact of the individual overexpression of the TFs essential for thyroid cell fate. When *Pax8* was over-expressed independently, the cells displayed limited differential capacity towards a TFC fate [31]. However, the *Pax8* knock-out (KO) mESC line did not differentiate into thyroid [72]. The authors speculate that Pax8 loss of function results in an inability of the cells to express high levels of endogenous *Nkx2-1*, which shows that in this model, *Pax8* is involved in the initiation of *Nkx2-1* expression [31]. On the contrary, when *Nkx2-1* was over-expressed independently, the induction was sufficient to promote a TFC-like state but failed to form follicles. Furthermore, the same mESC-derived thyroid model was also used to study the impact of Foxe1 on thyroid development. The authors reported reduced mRNA levels in many thyroid-related genes and a decrease in follicle formation coupled with the scarce detection of the sodium/iodide symporter, *Nis*. Additionally, *Foxe1* KO thyroid organoids showed an absence of Tg iodination and limited organification [72]. These in vitro results corroborated the in vivo phenotype in *FoxE1* null mouse embryos, which showed severe defects in thyroid morphogenesis [73]. In humans, homozygous mutations in the *FOXE1* loci result in congenital hypothyroidism or complete agenesis [74].

We also used mESC-derived thyroid organoids to study the CH-related mutations in the *Nrf2* gene. In this case, *Nrf2* KO organoids showed an absence of follicular organization, low *Tg* expression, and an inability to organify iodine and produce thyroid hormone, showing their usefulness as a model to study later developmental stages like follicle formation and hormone production [75].

Kurmann et al. (2015) also report the use of patient-derived iPSC lines from three children with brain-lung-thyroid syndrome with three different mutations in the *NKX2-1* coding sequence [30]. Upon differentiating these cell lines with *NKX2-1* haploinsufficiency, they report a rare occurrence of cells expressing *NKX2-1* and *PAX8* while observing a reduced expression of the thyroid differentiation markers [30]. These findings recapitulate the patient phenotype as this syndrome is characterized by reduced thyroid function at birth and fluctuations in TH synthesis [76]. A summary of these results is depicted in Figure 2.

mESC- and hESC-derived models have advanced the study of human development and disease but carry ethical concerns and lack genetic diversity [12,77]. Patient-derived iPSCs offer a promising alternative for disease modeling [78,79]. Human thyroid cells and organoids have been successfully generated from iPSCs, and future research on CH will likely focus on using patient-derived iPSCs, with or without specific mutations, to create thyroid organoids and study their phenotypes in vitro [30,36]. By leveraging directed differentiation protocols—though still requiring optimization for functionality—gene causality and molecular mechanisms can be thoroughly investigated. An alternative and highly efficient strategy involves the use of CRISPR/Cas9 technology to introduce patient-specific mutations into human PSCs. This approach allows the generation of functional thyroid organoids while providing the advantage of using an isogenic wild-type line as a control for precise comparisons. This outlook facilitates the study of the potential mechanisms underlying thyroid cell specification, survival, and organization over time, revealing insights that are challenging to obtain in vivo. Thyroid organoid technology is currently focused on restoring organ function in cases of the complete absence of the thyroid gland. However, another possible area of investigation could be the reactivation of low or excessive-producing tissue, for example in adult hypothyroidism and hyperthyroidism. The generation of patient-derived thyroid organoids, using tissue biopsies, reflecting the patients’ phenotype would permit the comparison to healthy and functional thyroid tissue and consequently identify the potential causative molecular pathways. This method could allow researchers to determine the disease mechanisms and potentially develop new treatments aiming to recover the disrupted thyroid tissue to a physiological state.

### 3.2. Organoid Modeling of Thyroid Autoimmune Diseases

Contrary to CHTD, in developing countries, the most common cause of primary hypothyroidism is not a lack of iodine but chronic autoimmune diseases. There are two types of thyroid autoimmune disorders: Graves’ disease and Hashimoto’s thyroiditis [80]. Graves’ hyperthyroidism is described as the overstimulation of the TSH receptor by TSH-mimicking antibodies resulting in the overproduction of THs. On the other hand, Hashimoto’s thyroiditis is characterized by an underactive thyroid coming from the failure of T cell-mediated inflammatory responses involving antigen-presenting T and B cells. Lymphocyte infiltration in thyroid tissue directly hampers follicular function and results in the underproduction of THs [81].

Using a tissue-derived organoid model, Van der Vaart et al. (2021) developed an in vitro system to model Graves’ disease [20]. They described thyroid organoids to be responsive to a panel of Graves’ disease patient sera and displayed a significant increase in growth and hormone production. Thus, the sera were triggering thyroid follicular cell proliferation and hormone secretion, recapitulating Graves’ characteristics in vitro. Despite the response and indicative phenotype, this model lacked the immune compartment to recapitulate the auto-immune condition. One possible approach to model the immune component is to co-culture thyroid organoids with immune cells, such as peripheral blood mononuclear cells (PBMCs) or isolated lymphocytes. Alternatively, human thyroid samples from autoimmune conditions, for example from Fine-Needle Aspiration (FNA) or surgically removed tissue-derived organoids could be generated and used to study disease mechanisms.

### 3.3. Using Thyroid Organoids to Study Thyroid Disruption by EDCs

Endocrine-disrupting chemicals (EDCs) are exogenous substances, either natural or synthetic, that interfere with the normal endocrine system function. They can affect hormone biosynthesis, release, transport, and degradation, or disrupt receptor signaling, acting as agonists or antagonists. These disruptions can affect health and the environment [82]. Over the years, numerous substances encountered in daily life have been proposed as EDCs. As a result, several EDCs have been banned or restricted in food-contact materials and drinking water applications. Health authorities have also issued precautionary recommendations to minimize the exposure to other potential EDCs in these contexts. Endocrine-disrupting chemicals can be classified into different categories, such as polycyclic aromatic hydrocarbons, heavy metals, phthalates, organophosphate flame retardants, and polychlorinated bisphenyls, based on their chemical nature, but many compounds are still not classified [83]. It is speculated that endocrine disruptors function through various mechanisms, including nuclear receptors, nonnuclear steroid hormone receptors (such as membrane estrogen receptors), nonsteroid receptors (like neurotransmitter receptors), orphan receptors (e.g., the aryl hydrocarbon receptor), and enzymatic pathways related to steroid biosynthesis and metabolism [84]. In the case of thyroid function, different EDCs were shown to impair the hypothalamus–hypophysis–thyroid (HHT) axis at different levels [85].

The use of organoids represents a significant advancement in studying the effects of EDCs on human health. Studies on the effects of EDCs in thyroid organoids are largely based on the mESC model developed by Antonica et al. (2012) [31]. Recent research introduced a thyroid organ-on-a-chip (OoC) device, offering a more physiological alternative to evaluate the impact of EDCs on thyroid function [55]. Interestingly, the exposure of thyroid organoids for 24 h to benzo[k]fluoranthene (BKF) revealed the activation of the aryl hydrocarbon receptor pathway, showcasing its ability to detect molecular responses to EDCs, and demonstrating the potential of the model for broader applications. This system enables the maintenance of organoids for extended periods, making it a promising platform for studying the long-term effects of EDC exposure. Even in the short-term, BKF significantly altered the expression of key pathways, highlighting the sensitivity of the model and its potential for future applications [55].

Using the same system, the effects of biologically relevant doses of various phthalates on thyroid transcriptome were evaluated [86]. Phthalates were added to differentiation media for 24 h, and transcriptional changes in thyroid organoids were analyzed via RNA sequencing. While thyroid differentiation gene expression remained unchanged, differential expression analysis revealed significant alterations in signal transduction and extracellular matrix organization. Chromatin accessibility was assessed using ATAC-seq, but no major significant differences were found. This study highlights that rapid transcriptional responses can be captured within 24 h of exposure while chromatin accessibility changes, dedifferentiation, and function, influenced by complex epigenetic processes, may require longer treatments to be detectable.

In a related study, Nazzari et al. (2024) utilized human thyroid organoids to investigate the effects of EDCs [87]. The study examined the sex-specific responses to EDCs by incorporating hormone cocktails into the differentiation media to simulate post-ovulatory conditions for females and reproductive-age levels for males. Thyroid organoids were treated with benzo[a]pyrene or PCB153 and analyzed via single-cell RNA sequencing. The results showed compound- and hormonal-variation specific effects, with benzo[a]pyrene reducing the *TG* expression in a “male” hormonal environment but not in a “female” one. The authors concluded that thyroid organoids are a reliable model for studying EDCs and advancing the related research.

The research on thyroid organoids for studying EDCs is still in its early stages, with a few successful proof-of-concept studies demonstrating the versatility of the model. While the short exposure durations may have limited the detection of long-term impacts, the sensitivity of the techniques used revealed important alterations, emphasizing the utility of thyroid organoids in this field.

Together, these studies demonstrate that these thyroid organoids are suitable as an in vitro model to study not only the effect of environmental exposure to toxins but also the basis of thyroid disease and development.

## 4. Challenges and Limitations in Using Thyroid Organoids for Therapeutic Applications

Over the last decade, several attempts have been made to generate functional thyroid organoids in the most physiological manner aiming to implement cell therapy for treating thyroid disorders, such as hypothyroidism. Thyroid hormone replacement therapy is the standard treatment for hypothyroidism [88]. However, precise dosage optimization remains challenging, requiring lifelong hormone supplementation [89,90,91]. Children are the primary population affected by hypothyroidism, experiencing complications that impact motor, cognitive, and social development [92]. During growth and puberty, they clinically present fluctuations in the levels of hormones, a similar situation observed during pregnancy. Additionally, movement, cognitive, and social disorders have been observed in children with CH, even when treated early and diagnosed promptly. This indicates that a proportion of patients could benefit from the use of personalized thyroid organoids as a therapy to restore thyroid function. Here, we discuss several challenges, from cell culture methods to transplantation, that must be overcome before adopting these organoids in a clinical setting.

There are still several challenges that must be addressed before using thyroid organoids models for clinical application [93]. One major limitation is that most protocols generate organoids with low efficiency or based on genetic modifications. Genome modifications might lead to off-target effects and adverse consequences, such as tumorigenesis, and their long-term effects are still unknown. To overcome such limitations, new tools should be developed to improve the efficiency of producing human iPSC patient-derived unmodified thyrocytes, such as fully automated bioreactor systems [94]. Additionally, the current efficiency and functional protocols are based on genetic modifications needed to enable exogenous transcription factor overexpression for cell specification and to create transgenic cell lines for tracking and purification. A promising approach is the generation of functional organoids by directed differentiation and the identification of surrogate cell surface markers allowing the selection and purification of thyroid progenitors without a knock-in reporter construct. Also, for both in vitro and in vivo applications, organoid culture should rely on matrixes from non-animal origins to ensure the safety of the transplantation. Testing biocompatible materials for organoids’ encapsulation (alginate, agarose, cellulose, etc.) can optimize the cell preservation and immune isolation of transplanted cells while allowing proper nutrient exchange and hormone secretion. Furthermore, fetal bovine serum (FBS) is commonly used in organoid cultures to achieve differentiation and maturation, but every batch varies in composition, making standardization challenging. This inconsistency affects differentiation, proliferation, and maturation while introducing unknown factors that can cause batch-to-batch variability. Standardized, chemically defined serum replacements must be identified to improve reproducibility. Another major challenge is that these directed differentiation models show a low level of maturation, heterogeneous population, batch-to-batch variability [37,95], and genetic modifications in the cells, which can further impact the differentiation efficiency and reproducibility [94]. Good Manufacturing Practices (GMPs) and controlled differentiation protocols are potentially key factors to reach this aim.

The immune rejection of transplanted organoids by the host is a significant barrier to adopting organoids in the clinical fields. Some of the potential solutions to reduce the immunogenicity after transplantation are the use of iPSCs starting from patient-derived cells, the correction of mutations that might lead to immunogenicity, deletion of the human leucocyte antigen (HLA) genes to block the immune reaction, and adoption of immune modulatory molecules to avoid an immune reaction response [96,97,98].

Another consideration is to determine the optimal number of transplanted follicles or cells to restore the concentration of TH levels. It varies from person to person based on gender, age, weight, and height. Further studies are essential to determine the quantity–response relationships and permit an appropriate hormonal secretion. Similarly, the transplantation site for functional assessment in mice must be safe and allow the detection of hormone production, induce a reduced immune tolerance, permit the study of the tissue, and easily permit vascularization. Thus far, the kidney capsule has been used for testing the in vivo functionality of transplanted thyroid organoids; however, a more accessible location reducing the complexity of the surgical procedure should be tested. Finally, the assessment of long-term safety and functional effects must be performed to detect or identify the potential changes in the TH levels.

The difficulties in assessing the responsiveness of the different tissues to synthetic thyroid hormones outline the relevance of focusing on the regeneration strategy as the major technique to adopt in the clinic. This, together with finding suitable methods that allow a constant and non-invasive detection of the presence of THs in serum, is important. Overall, all the above-cited research and the constant progress in thyroid research outline that cell therapy holds relevant promise as a potential treatment for thyroid disorders and to improve patients’ quality of life, providing new strategies both for preclinical and clinical studies.

## 5. Conclusions

The field of thyroid organoids has made significant progress from its birth in understanding thyroid development and function. Thyroid organoids serve as a versatile platform for investigating the developmental mechanisms, modeling diseases, studying hormone synthesis regulation, conducting drug screening, and assessing toxicity. Efforts are underway to optimize the protocols for generating fully functional thyroid organoids in both in vivo and in vitro settings. Advances include serum-free protocols and the use of iPSC-derived models, which hold the potential for clinical applications but are currently limited by incomplete maturation and reduced functionality post-transplantation. Innovative approaches, such as co-culture systems, microfluidics, organ-on-a-chip, and bioprinting, are emerging to replicate the physiological environment of the developing thyroid or model disease states, such as autoimmune disorders. Despite these advances, challenges remain in ensuring safety and optimizing the transplantation conditions. The field shows promise in reducing the reliance on animal models, creating stable, functional human thyroid tissue, and potentially offering alternatives to lifelong hormone replacement for patients with thyroid dysfunction.

## Figures and Tables

**Figure 1 biomolecules-15-00747-f001:**
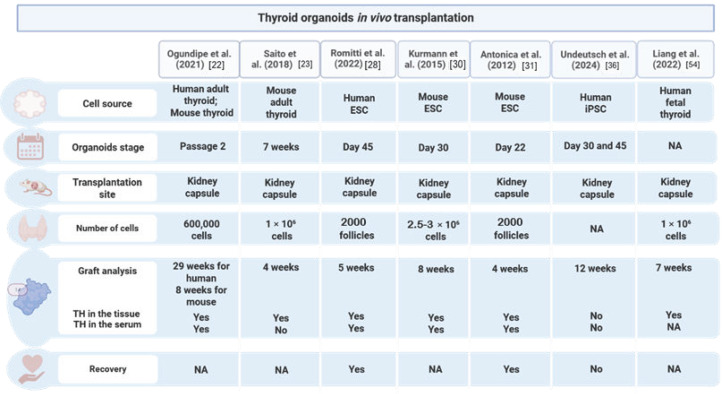
Summary of in vivo transplantation studies, with a focus on the organoid stage, site of transplantation, number of cells, graft analysis, TH quantification, and hypothyroidism rescue. NA: not available.

**Figure 2 biomolecules-15-00747-f002:**
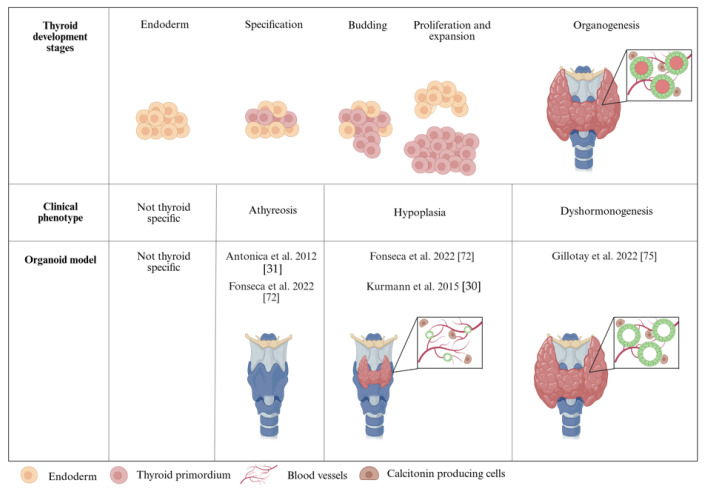
Thyroid organoids are used to model congenital hypothyroidism and the phenotypes observed at different developmental stages.

**Table 1 biomolecules-15-00747-t001:** Summary of studies that have generated thyroid organoids from PSCs.

Study	Cell Source	Type of Differentiation	Specification Efficiency	Total Efficiency	Maturation In Vitro	In Vitro Functionality		Transplantation		Reference
						T4 Detection	Tg-I Detection	T4 + Tissue	Systematic Recovery	
Romitti et al., 2022	human-ESCs	Forward programing	~12%	~25%	Full	Yes	Yes	Yes	Yes	[28]
Longmire et al., 2012	mouse-ESCs	Directed differentiation *	16%	~60%	Partial	NA	NA	NA	NA	[29]
Kurmann et al., 2015	mouse-ESCs	Directed differentiation *	4.90%	50%	Partial	No	NA	Yes	Yes	[30]
Antonica et al., 2012	mouse-ESCs	Forward programing	NA	60.5 ± 8.1%	Full	No	Yes	Yes	Yes	[31]
Ma et al., 2013	mouse-ESCs	Forward programing	NA	NA	Partial	NA	NA	NA	NA	[32]
Ma et al., 2015a	mouse-iPSCs	Forward programing	NA	NA	Partial	NA	NA	NA	NA	[33]
Ma et al., 2015b	human-ESCs	Forward programing	NA	NA	Partial	NA	NA	NA	NA	[34]
Dame et al., 2017	mouse-ESCs	Forward programing **	~5%	NA	Full	Yes	NA	No	No	[35]
Undeutsch et al., 2024	human-iPSCs	Directed differentiation *	35%	73.90%	Partial	No	NA	No	No	[36]
Ma et al., 2020	human-iPSCs	Forward programing	34.29%	58.35%	Full	Yes	NA	NA	NA	[37]

NA: not available; * thyroid progenitors’ enrichment; ** enriched at anterior foregut endoderm stage.

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
