# Peer review of "Functional Thyroid Organoids—Powerful Stem Cell-Derived Models in Basic and Translational Research"

_biomolecules, 2025, doi:10.3390/biom15050747_

Round 1
Reviewer 1 Report
Comments and Suggestions for Authors
The one comment I have is in line 316 it is stated that mESC-derived orgnaoids are used in references 22. However these are adult stem cells derived from tissue organoids that are transplanted.
Further I find it a pity that cancer treatment is not mentioned as a way hypothyroidism is caused. Moreover, also thyroid cancer organoids have been cultured and use to study TC.I my opinion this is an omission that should be added.
Author Response
Comment 1: The one comment I have is in line 316 it is stated that mESC-derived orgnaoids are used in references 22. However, these are adult stem cells derived from tissue organoids that are transplanted.
Response 1: Dear reviewer, thank you for pointing it out. We apologize for the error on our part. The sentence has been corrected (Line 336).
Comment 2: Further I find it a pity that cancer treatment is not mentioned as a way hypothyroidism is caused. Moreover, also thyroid cancer organoids have been cultured and use to study TC.I my opinion this is an omission that should be added.
Response 2: Dear reviewer, following your suggestion we have added a few sentences to clarify the possible causes of reduced thyroid function, including the surgical ablation for thyroid cancer (L.380-382). In addition, we would also like to clarify that the review being part of a special issue Biosynthesis and Function of Thyroid Hormones, the scope proposed was focusing on Thyroid function and did not include TC organoids . Here, we stated the absence of TC organoids and we have referred to the recent reviews on the topic: Prete et. al. 2024, titled ‘State of the Art in 3D Culture Models Applied to Thyroid Cancer. Medicina’ and Song et. al. 2024 entitled ‘Advances in Thyroid Organoids Research and Applications. Endocr Res’ on the subject of thyroid cancer. Please see L.386.
Reviewer 2 Report
Comments and Suggestions for Authors
In this review article, the authors provide a concise overview of stem cell-derived models which can be used as functional thyroid organoids. They highlight the applications of such thyroid organoids for understanding thyroid development including thyroid hormone biosynthesis, and modeling the thyroid health and diseases. The challenges and limitations of using thyroid organoids for therapeutic applications are also discussed. Overall, this is an interesting review on the potential applications of thyroid organoids in both basic research and translational aspects. The paper is suitable for publication in Biomolecules after minor revision as given below.
1. The authors discuss about the thyroid hormone synthesis in organoids. It would be helpful if the authors can describe the relative levels of thyroid peroxidase (TPO), ratio of T4/T3 generation, the presence of thyroid hormone transporters in different organoids. Can the organoid models be used for the synthesis of thyroid hormone analogues or thyromimetics?
2. The authors may discuss about the application of organoids in the development of novel therapeutics for thyroid disorders such as hypothyroidism and hyperthyroidism by modulating the thyroid hormone biosynthesis pathways.
3. The authors mention that the thyroid organoids can be used to study disruption of thyroid by endocrine disruptor chemicals (EDCs). It would be helpful to the readers if the authors can provide the chemical structures or class of some of these EDCs which can be studied using organoids.
Author Response
Comment 1: In this review article, the authors provide a concise overview of stem cell-derived models which can be used as functional thyroid organoids. They highlight the applications of such thyroid organoids for understanding thyroid development including thyroid hormone biosynthesis, and modeling the thyroid health and diseases. The challenges and limitations of using thyroid organoids for therapeutic applications are also discussed. Overall, this is an interesting review on the potential applications of thyroid organoids in both basic research and translational aspects. The paper is suitable for publication in Biomolecules after minor revision as given below.
Response 1: Dear reviewer, Thank you for your valuable comments. We have thoroughly revised the manuscript to enhance the clarity, grammar, and overall readability of the English language.
Comment 2: The authors discuss about the thyroid hormone synthesis in organoids. It would be helpful if the authors can describe the relative levels of thyroid peroxidase (TPO), ratio of T4/T3 generation,the presence of thyroid hormone transporters in different organoids.
Response 2: We appreciate the reviewer's insightful suggestion regarding the role of various factors in organoids' thyroid hormone production. In this revised version, we have thoroughly evaluated the available data from both in vivo and in vitro studies and discussed potential experiments to elucidate the impact of replacing native thyroid tissue with organoids on body homeostasis. We also emphasize the variability in the methods used, which complicates the comparison of studies and the drawing of conclusions. The lack of standardized evaluation methods underscores the need for further research in this area. All changes are highlighted in the following sections: 2.4. Thyroid hormone synthesis (L.242-301) and 2.5. In vivo transplantation and rescue of hypothyroidism (L.302-368).
Comment 3: Can the organoid models be used for the synthesis of thyroid hormone analogues or thyromimetics?
Response 3: Dear reviewer, While the prospect is intriguing, we believe that thyroid organoids generated in vitro, like native thyroid tissue, can only synthesize endogenous thyroid hormones, not synthetic analogs. However, we are open to discussing other perspectives that we may not have considered.
Comment 4: The authors may discuss about the application of organoids in the development of novel therapeutics for thyroid disorders such as hypothyroidism and hyperthyroidism by modulating the thyroid hormone biosynthesis pathways.
Response 4: Dear Reviewer, thank you for raising these previously unaddressed perspectives. This is indeed a novel application of in vitro organoids with significant potential. Currently, the focus of generated organoids is on restoring organ function through transplantation, particularly in conditions with a complete absence of thyroid tissue, such as congenital hypothyroidism (CH), hyperthyroidism, or thyroid cancer (TC) treatment. Exploring ways to ‘reactivate’ low-producing tissues could be investigated in vitro by examining which pathways might be modulated to enhance thyroid hormone (TH) production. For example, in the case of hypothyroidism and hyperthyroidism, we could mimic the patient phenotype in vitro, as demonstrated for Graves’ disease by Van der Vaart in 2021 using adult thyroid tissue-derived organoids. Additionally, by comparing the transcriptomics of healthy thyroid organoids with those derived from hyperthyroid or hypothyroid conditions, we could identify potential mechanisms and contribute to the development of new treatment strategies for these specific cases. We have discussed these potential perspectives in section 3.1.1 (L.466-474).
Comment 5: The authors mention that thyroid organoids can be used to study disruption of thyroid by endocrine disruptor chemicals (EDCs). It would be helpful to the readers if the authors can provide the chemical structures or class of some of these EDCs which can be studied using organoids.
Response 5: Dear Reviewer, thank you for your assistance in making this review more comprehensive. Due to the limited length of the review, we chose to describe the main classes of endocrine-disrupting chemicals (EDCs) and discuss the proposed mechanisms by which they could impact thyroid function (L.505-513). We also referenced previously published works that detail the classification, mode of action, and specific EDCs shown to influence thyroid tissue homeostasis.
Reviewer 3 Report
Comments and Suggestions for Authors
The work by Meghna Parakkal Shankar et al., entitled “Functional thyroid organoids – powerful stem cell-derived models in basic and translational research,” presents a literature review on the development, characterization, and potential applications of functional thyroid organoids.
The manuscript is well-written, presented in an intelligible and logical manner, and the language is clear and appropriate. The figure and table included are informative and visually clear. Most of the references are up-to-date and relevant to the topic.
As a minor comment, the authors are encouraged to include a brief paragraph describing the methodology used for the literature search. Details such as the databases searched, time frame covered, and inclusion/exclusion criteria would enhance the transparency and reproducibility of the review.
Author Response
Comment 1: The work by Meghna Parakkal Shankar et al., entitled “Functional thyroid organoids – powerful stem cell-derived models in basic and translational research,” presents a literature review on the development, characterization, and potential applications of functional thyroid organoids.
The manuscript is well-written, presented in an intelligible and logical manner, and the language is clear and appropriate. The figure and table included are informative and visually clear. Most of the references are up-to-date and relevant to the topic.
As a minor comment, the authors are encouraged to include a brief paragraph describing the methodology used for the literature search. Details such as the databases searched, time frame covered, and inclusion/exclusion criteria would enhance the transparency and reproducibility of the review
Response 1: Dear Reviewer, thank you for the positive comments on our review and for highlighting this important methodological aspect. We have included the search and selection criteria used to identify the articles discussed in this review in section L.94-102.